# Septic Two-Stage Cementless Hip Revision Arthroplasty Is Safe but Has Higher Complication and Mortality Rates in Older Adults

**DOI:** 10.3390/jcm14186556

**Published:** 2025-09-18

**Authors:** Florian Hubert Sax, Marius Hoyka, Benedikt Paul Blersch, Leonard Grünwald, Bernd Fink

**Affiliations:** 1Department for Joint Replacement, Rheumatoid and General Orthopaedics, Orthopaedic Clinic Markgröningen, Kurt-Lindemann-Weg 10, 71706 Markgröningen, Germany; florian.sax@rkh-gesundheit.de (F.H.S.); marius.hoyka@rkh-gesundheit.de (M.H.); benedikt.blersch@rkh-gesundheit.de (B.P.B.); leonard.gruenwald@rkh-gesundheit.de (L.G.); 2Department of Trauma and Reconstructive Surgery, BG Klinik, University of Tübingen, Schnarrenbergstraße 95, 72076 Tübingen, Germany; 3Orthopaedic Department, University Hospital Hamburg-Eppendorf, Martinistrasse 52, 20251 Hamburg, Germany

**Keywords:** geriatric patients, septic two-stage revision, hip revision arthroplasty, transfemoral approach, extended trochanteric osteotomy, periprosthetic infection

## Abstract

**Background:** Two-stage septic hip revision arthroplasty has higher mortality rates than aseptic hip revision arthroplasty, and patients over 75 years have higher fracture rates than younger patients after cementless total hip arthroplasty. Therefore, the aim of this study was to examine whether two-stage septic hip revision arthroplasty in older patients leads to higher complication and mortality rates, as well as whether changing to cementless hip prostheses in older patients could lead to higher fracture and subsidence rates of the stem prosthesis than in younger patients. **Material and Methods:** In total, 286 two-stage-revision procedures for periprosthetic infections of the hip in 186 patients younger and 118 patients older than 75 years were followed for a minimum of 24 (50.24 ± 20.77) months. A total of 71.3% of procedures were performed via a transfemoral approach using cementless cups and revision stems (93.7%). Complications and mortality were analyzed retrospectively. **Results:** There was a one-year mortality rate of 1.0% with no difference in the groups, and a general mortality rate of 2.8% with a significantly higher rate in older adults than in the younger group (5.9% vs. 0.6%; *p* = 0.01). The rate of fractures of the bony flap in transfemoral approaches (9.1%), fissure rate of the isthmus (2.8%), rate of subsidence of cementless stems (1.0%), and rate of reinfections (4.89%) did not differ between the two groups. The general complication rate (not associated with cementless two-stage septic revision) (22.0%) was significantly higher in the older patient group (33.9% vs. 13.7%; *p* < 0.001). **Conclusions:** Septic two-stage revision hip arthroplasty, mostly using a transfemoral approach and cementless reimplantation, does not result in a higher one-year mortality rate, reinfection rate, and rate of fissures and fractures of the bony flap, but demonstrates a generally higher mortality and complication rate in older adults. This should be taken into consideration when determining the indication and when offering advice to older adults.

## 1. Introduction

Periprosthetic infections are serious complications after total hip replacements and occur in approximately 1 to 2% of cases [1]. In the case of late infections (later than 4 to 6 weeks after implantation), the implants usually have to be removed as the bacteria have formed a biofilm on the implant that is difficult for antibiotics and immune defense cells to penetrate [2]. A distinction is made here between a one-stage procedure (with direct implantation of a new, usually cemented prosthesis) and a two-stage procedure (usually with temporary implantation of a spacer and delayed reimplantation of a new prosthesis) [2]. The two concepts have comparable success rates in terms of freedom from infection, as prosthesis registers have shown [3,4]. Arguments in favor of a two-stage exchange include the possibility of debridement twice and cementless reimplantation, because the interdigitation of the cement is not effective due to the mostly smooth walls of the femur, and the fixation of the new implant could therefore be inferior [2,5]. Some authors therefore favor a septic two-stage exchange over cementless implants in the hip joint [2,6,7].

On the other hand, for primary total hip arthroplasty, it is evident that cementless stem fixation has significantly more complications than cemented stem fixation in older adults [8,9]. Periprosthetic fracture is the leading mode of failure for cementless stem fixation in these older adults, and decreased bone quality is the reason for that [8,9,10]. The age limit between these groups was set at 75 years in many studies [8,11]. In older patients, septic two-stage revision to cementless prosthetic components may therefore lead to greater fracture rates, while the need for two operations may lead to higher rates of general complications and mortality rates. In general, higher complication and slightly higher mortality rates were found for two-stage septic revisions compared to single-stage septic prosthesis exchange [12] and significantly higher mortality rates for septic hip prosthesis revisions than for aseptic prosthesis exchanges [13,14].

Therefore, the aim of this study was to examine whether two-stage septic prosthesis exchanges in older patients lead to higher complication and mortality rates and whether switching to cementless hip prostheses in older patients leads to higher fracture and subsidence rates of the prosthesis compared to younger patients.

## 2. Materials and Methods

This study was designed as a single-center, retrospective analysis. Inclusion criteria comprised patients with confirmed chronic periprosthetic joint infection (PJI) of the hip between 2017 and 2023, defined according to the International Consensus Meeting (ICM-criteria 2018) [15], who underwent a two-stage revision procedure. Exclusion criteria were cases in which no periprosthetic infection was present, patients who had undergone septic one-stage procedures (*n* = 55), patients with acute infections treated with prosthesis-retaining procedures (*n* = 65), and patients with a follow-up of less than 24 months (*n* = 12). Between 2017 and 2023, 298 two-stage revision procedures for periprosthetic infections of the hip were performed at our institution. In total, 12 patients had a follow-up of less than 24 months, leaving 286 patients.

Preoperatively, all patients underwent joint aspiration with synovial fluid analysis, including microbiological cultures in blood culture bottles, determination of synovial alpha-defensin, and leukocyte count. In cases of inconclusive findings or dry taps, a minimally invasive biopsy was performed with the collection of five tissue samples for microbiological analysis and five samples for histological analysis. The microorganisms were divided into categories of easy-to-treat (EET), methicillin-resistant (MRS), difficult-to-treat (DTT), and culture-negative (CN, without detection of a germ) according to Faschingbauer et al. [16].

During the first-stage procedure, the infected implants were removed, and a cemented antibiotic-loaded spacer was implanted (104 times using an endofemoral approach and 182 times using a transfemoral approach) (Figure 1). The local antibiotic admixture was selected according to the antibiogram, as described previously [2,6]. The most frequently used antibiotics were gentamicin, clindamycin, and vancomycin, which were consistently applied in combination to optimize local bioavailability and improve elution characteristics. Radical debridement was performed, and a local antiseptic solution was applied. Additional intraoperative samples for microbiology and histology were obtained. Postoperatively, patients received a standardized antibiotic regimen consisting of 2 weeks of intravenous therapy followed by 4 weeks of oral antibiotics according to the antibiogram of the microorganisms.

At the time of reimplantation, the spacer was removed, followed by thorough debridement, application of a local antiseptic solution, and reimplantation of a new prosthesis (Figure 1). Again, patients received 2 weeks of intravenous antibiotics followed by 4 weeks of oral therapy, according to the susceptibility of microorganisms.

At the second stage, reimplantation was performed in 64 cases using the endofemoral approach in 60 cases with a cementless cup and cementless standard stem; in 4 cases, a cementless cup and a cementless modular tapered, fluted revision stem were used (Revitan curved, ZimmerBiomet, Winterthur, Switzerland). In the 182 cases with a transfemoral revision, the tapered, fluted, and modular revision stem was used, and 22 cases had a “transfemoral cementless fixation + distal locking”; In 15 cases, we performed surgery with “cemented fixation, and in 2 cases, surgery was performed with “prox. femoral replacement” and 1 “Girdlestone”.

In cases with transfemoral revision, cerclages were used for osteosynthesis of the bony flap of the osteotomy, as described previously [2,6,17,18].

The patients were followed prospectively for a minimum of 24 months and data were collected in our clinical data base, including patient demographics (age, sex, body mass index, comorbidities, ASA score, Charlson Comorbidity Index, diabetes mellitus, and rheumatic disease) and prosthesis-associated variables (duration of implant prior to diagnosis of infection and the extent of bone loss, classified according to the Paprosky classification system [19,20,21]). Outcome variables included eradication of infection at the latest follow-up (reinfection rates), the need for further revision surgery, and implant survival and complication rates (e.g., fracture, subsidence, and general complications). General complications were divided into ischemic (apoplexy, myocardial infarction), thromboembolic (deep vein thrombosis, pulmonary artery embolism), nephrological (renal insufficiency, urinary tract infection), pulmonal (pneumonia), cardial (cardiac decompensation, endocarditis), hemorrhagic (hematoma, gastrointestinal hemorrhage, subarachnoid hemorrhage) and soft tissue complications (soft tissue defect, wound healing disorder, necrosis, erysipelas). Retrospective data analysis was performed. The follow-up time period was on average 50.24 ± 20.77 months (time span 24–95 months).

Statistical analysis was conducted using IBM SPSS Statistics for Windows (version 24, IBM Corp., Armonk, NY, USA). To calculate group differences, univariate analysis of variance was used in the case of metric variables and chi-squared tests, as well as Mann–Whitney U tests for variables on an interval or nominal-scale level. To check if potential independent predictors can be identified, independently of their age as a dichotomizing variable, logistic regression analysis was performed. Unless otherwise stated, data are presented as the mean ± standard deviation or number (percentage). The significance level was set at *p* < 0.05. Informed consent was obtained from all participants, and the study protocol was approved by the research ethics boards of the Landesärztekammer Nord-Württemberg (F-2023-115).

## 3. Results

The mean patient age was 70.7 ± 11.2 (25.4–92.8) years, with 161 males (56.3%) and 125 females (43.7%). The body mass index (BMI) of the patient cohort averaged 29.8 ± 6.8 (16.9–60.6) kg/m^2^. Results from the American Society of Anesthesiologists (ASA) risk classification as well as the Charlson Comorbidity Index (CCI) can be seen in Table 1. In terms of secondary diseases, 13 patients (4.5%) had rheumatic disease, and 66 patients (23.1%) had diabetes mellitus. Septic revision was carried out in 95 patients (33.2%) prior to the analyzed two-stage septic revision.

The average time between the primary implantation and the analyzed two-stage septic revision was 99.2 ± 80.78 (1–500) months. Demographic data is shown in Table 1.

The total study sample was divided into two age groups: group 1 for patients younger than 75 years (25.4–74.8 years; *N* = 168; 58.7%) and group 2 for patients older than 75 years (75.1–92.8 years, *N* = 118; 41.3%). Whereas gender distribution was almost equal between groups 1 and 2 for females (52% in group 1; 48% group 2), for males, gender distribution changed with age (64% in group 1 vs. 36% in group 2); these differences reached statistical significance (χ^2^(1) = 4.164; *p* = 0.041). The average age as well as BMI also differed significantly between groups (*p* < 0.001). The Paprosky classification differed significantly between age groups as well (χ^2^(4) = 14.852; *p* = 0.005). The median score was 2.5 for the whole study population. Patient characteristics are shown in Table 1.

Significant differences (χ^2^-testing, both *p* < 0.001) could also be observed for the American Society of Anesthesiologists (ASA) risk classification and the Charlson Comorbidity Index (CCI) between groups 1 and 2 (Table 1).

The surgical techniques employed differed significantly between age groups (*p*< 0.001; Table 2), as a greater number of patients in the subgroup of older adults received a transfemoral cementless fixation (72.9%).

Highly significant differences in surgical technique also occurred between groups derived from the Paprosky classification (*p* < 0.001): 97.3% of all patients grouped in IIIa received surgery performed with “transfemoral cementless fixation”, whereas almost all patients with classifications of IIIB and IV received surgery with “transfemoral cementless fixation + distal locking” (except one patient IIIB who received “transfemoral cementless fixation”).

Regarding analysis of microbiology and histology, in the total sample for 30 patients, germs were not detectable, whereas 177 patients had an easy-to-treat microbiology, 21 patients had a difficult-to-treat microbiology, and 58 patients showed evidence for methicillin-resistant Staphylococcus (MRS). In the statistical analysis performed, no difference was observed between age groups; however, a trend was detectable, showing that in the older group, often, no microbial detection was possible. The results are displayed in Table 3. No association could be detected between the type of germ and risk of reinfection (χ^2^(4) = 2.060; *p* = 0.725). Concerning subsidence, all three observed cases had a microbiological classification of easy to treat. Due to the low number of cases with subsidence, no statistical significance was evident (χ^2^(4) = 1.895; *p* = 0.755).

The one-year mortality rate was 1.04% (3 of 286) and did not differ significantly between groups (*p* = 0.370): two patients (1.7%) of the older subgroup died within one year after surgery (age range 80.8–85.3), whereas one patient of the younger subgroup died (0.6%) (age 59.8). Post hoc power calculations resulted in a power of 16.4%. The general mortality rate was 2,8% (8 out of 286 patients) and differed significantly between groups (*p* = 0.010): seven patients (5.9%) of the older subgroup died (age range 86.9–78.5), whereas 1 patient of the younger subgroup died (0.6%) (age 59.8). Reinfection rates did not differ significantly between groups, with 9 reinfections in group 1 (5.4%) and 5 reinfections in group 2 (4.2%) (*p* = 0.445) (Table 4).

A total of 16 patients (9.5%) in group 1 and 10 patients (8.5%) in group 2 experienced a fracture of the bony flap during surgery in cases using a transfemoral approach, although the difference between groups did not reach statistical significance (*p* = 0.783). Twenty-four fractures were treated with cerclage during closure of the flap, and in two cases with an additional claw plate. A fissure at the isthmus of the femur was observed in six cases in group 1 (3.6%) and in two cases in group 2 (1.7%). These were treated with an additional distal double cerclage. The number of fissures was not significantly different between age groups (*p* = 0.286). Furthermore, a significant connection between the number of fissures and Paprosky classification [18,19,20] could not be observed (*p* = 0.883). Only three patients of the total study population showed a subsidence, all three of which belonged to group 1 (younger patients) (Table 4). However, the rate of subsidence did not differ significantly between age groups (*p* = 0.201).

Concerning general complications (not associated with cementless septic two-stage revision and independent of any type of complication), a significant difference was observed between age groups (*p* < 0.001): a total of 23 patients (13.7%) in the younger subgroup experienced complications, whereas a total of 40 patients (33.9%) experienced complications in the older subgroup. Table 5 gives an overview of the complications experienced in both age groups.

To check if potential independent predictors could be identified—independently of age as a dichotomizing variable—we performed logistic regression analysis with the following variables as independent variables: one-year mortality, reinfection rate, general complication rate, and subsidence. For all logistic regression analyses, the statistical criteria for the performance of a logistic regression were met. Linearity was tested and assessed using the Box-Tidwell [22] procedure. Bonferroni correction was applied to all terms in the model. All variables were found to follow a linear relationship. Correlations between predictor variables were low (r < 0.70), indicating that multicollinearity was not a confounding factor in the analysis. However, for the variables of one-year mortality and subsidence, no significant predictors could be found. Independent variables were the reinfection rate, age, the Paprosky index, ASA, CCI, BMI, diabetes, and gender.

For the variable reinfection, a reduced analysis was performed where only the predictors gender, age, BMI, the Paprosky index, and diabetes were assessed, and a significant model returned, resulting in an acceptable amount of explained variance with χ^2^(8) = 17.439, *p* = 0.026, and a Nagelkerke’s R^2^ = 0.202 [23]. The variable BMI was the only variable that contributed significantly to the model (*p* < 0.001).

For general complications, a reduced analysis with only the predictors gender, age, BMI, the Paprosky index, and diabetes was performed. The binomial logistic regression model was statistically significant, χ^2^(8) = 27.813, *p* < 0.001; however, this only resulted in a low amount of explained variance, as shown by Nagelkerke’s R^2^ = 0.129. The variables of age (*p* = 0.016) and diabetes (*p* = 0.031) contributed significantly to the model.

## 4. Discussion

The one-year mortality rate of two-stage septic hip revision arthroplasty in our study was relatively low at 1.0% and was lower than that reported by Zmistowki et al. [15] (at 3.7%), the Australian registry (at 3.7%) [14], the meta-analysis by Natsuhara et al. [24] (at 4.2%), and significantly below the rate recorded in the Danish registry (at 7.6%) [25] and the U.S. Medicare data base at 11.3% in patients over 65 years of age [26]. This may be explained by the fact that our center is a tertiary referral center for revision arthroplasty with both surgical and multidisciplinary team experience and septic revision arthroplasty; surgeries were performed only by highly experienced surgeons. This may underline the importance of a multidisciplinary approach in a specialized referral center to optimize outcomes in such complex cases.

In contrast to the one-year mortality rate, the general mortality rate of 5.9% in patients over 75 years of age was significantly higher than that of 0.6% in patients under 75 years of age in our study. This may be due to the fact that the older group had significantly higher ASA scores of three and a Charlson Comorbidity Index (CCI) ranging from 6 to 10, which are likely to influence the mortality rate. It is also conceivable that preoperative malnutrition, which is more common in geriatric patients, can lead to a higher mortality rate. In general, higher perioperative complication rates for total hip arthroplasty revision are observed in geriatric patients in connection with preoperative malnutrition [27]. This, alongside the significantly higher CCI and ASA score of three, may explain the fact that the complication rate in our study was significantly higher in the older patient group.

The reinfection rates were similarly low in both age groups at around 5% and were, therefore, at the same level as in the study by Camurcu et al. [7] and slightly better than 8.4% in the meta-analysis by Goud et al. [3]. Patients with an advanced age therefore do not appear to have a higher risk of reinfection, and even two-stage septic hip revision arthroplasty leads to reproducibly good freedom from infection in older people.

A fracture of the bone flap occurred during surgery in approximately 9% of cases in both groups. This was due to infection-related osteolysis and weakening of the bone in this area. There were no uncontrolled fractures below the flap, only 2.8% fissures, and no significant difference between the two age groups. These were treated with additional cerclage wires. One reason for the absence of a higher rate of uncontrolled periprosthetic fractures in patients over 75 years of age may be that, in our study, the transfemoral approach was consistently chosen for bone conditions at risk of fracture. The transfemoral approach has been shown to be sufficient at preventing or reducing the occurrence of uncontrolled periprosthetic fractures [17,18].

A correlation between the occurrence of fissures and fractures could not be established with the existing bone defect situation. This may be due to the fact that patients who underwent a transfemoral approach to remove fixed femoral stems were classified as Paprosky 3A or 3B [19,20,21] (depending on the remaining fixation zone in the isthmus) without necessarily being characterized by poor bone quality. It can be assumed that the risk of developing a fissure in the isthmus of the femur depends on the cortical thickness in this area. However, this is not categorized in the classification system of Paprosky [19,20,21]. The subsidence rate of 1.0% was very low in our study and was only observed in younger patients. The low incidence may also result from the transfemoral approach being chosen in 71.3% of cases, which allows the distal anchoring of the revision stem in the isthmus of the femur to be reproducibly well controlled [17,18].

There are several weaknesses to this study. One weakness may be that the boundary between the two age groups was drawn at 75 years and not higher, which may obscure age-related distributions. The reason for this was to increase the size of the elder group (168 younger and 118 older than 75 years). In our study, complication rates were experienced in individuals no older than 80 years of age. Another weakness is the retrospective approach of this study. This can lead to a loss of patients during follow-up and cannot exclude the fact that the general mortality rate is higher than mentioned. Furthermore, due to the low number of deaths (n = 8), which may be explained by the expertise of our center as a tertiary referral hospital with both surgical and multidisciplinary team experience, the power of mortality analysis is limited. Finally, no clinical outcome scores (such as Harris Hip Score, OHS, and EQ-5D) were assessed in this study, as this was not the primary aim. These aspects may be addressed in future investigations to determine whether patients without complications achieve satisfactory functional outcomes, thereby providing evidence to avoid overtreatment.

In summary, it can be said that even in older patients over the age of 75, two-stage septic hip revision arthroplasty using the transfemoral approach and cementless implants leads to low one-year mortality rates and fissure rates in the isthmus of the femur, as well as similar reinfection rates in younger patients. However, the general mortality rate and complication rate of older patients are higher than those of younger patients, probably due to the more common concomitant diseases experienced by these patients. This should be taken into account when determining the type of surgery and providing information to patients.

## Figures and Tables

**Figure 1 jcm-14-06556-f001:**
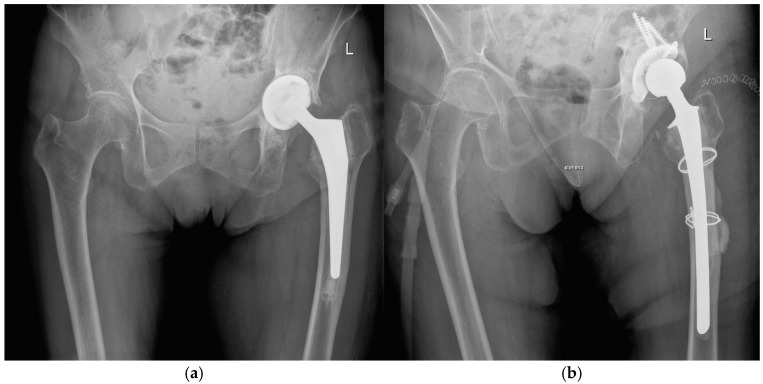
Radiographs of a two-stage procedure using a transfemoral approach performed on a 87-year-old women: (**a**) infected prosthesis with loosening of the cup and cemented stem; (**b**) temporary implanted spacer with cemented cup and cemented stem using a transfemoral approach; (**c**) reimplanted cementless prosthesis using the transfemoral approach; (**d**) one year follow-up of the reimplanted prosthesis with osseointegration of the components into the bone.

**Table 1 jcm-14-06556-t001:** Demographic data of the total study sample and the analyzed age groups. Statistically significant differences (*p* < 0.05) are marked (*).

Group 1 Younger 75	Group 2 Older 75	Total Sample	
*N*	*N*	*N*	
(%)	(%)	(%)	
63.46 ± 9.19	80.44 ± 3.51	70.47 ± 11.17	**Age *** (χ^2^, *p* < 0.001)
65 vs. 103	60 vs. 58	125 vs. 161	**Gender** **female vs. male *** (χ^2^, *p* = 0.041)
(38.7% vs. 61.3%)	(50.8% vs. 49.2%)	(43.7% vs. 56.3%)	
33	33	66	**Diabetes** (χ^2^, *p* = 0.067)
(19.6%)	(28.0%)	(23.1%)	
8	5	13	**Rheum. Dis.** (χ^2^, *p* = 0.537)
(4.8%)	(4.2%)	(4.5%)	
31.11 ± 7.08	27.84 ± 5.96	29.76 ± 6.83	**BMI *** (χ^2^, *p* < 0.001)
**Median score:** **2.0**	**Median score:** **2.0**	**Median score:** **2.0**	**Paprosky *** (χ^2^, *p* = 0.005)
5	2	7	I
(71.4%)	(28.6%)	(2.4%)	
52	16	68	II
(76.5%)	(23.5%)	(23.8%)	
98	84	182	IIIa
(53.8%)	(46.2%)	(63.6%)	
8	13	21	IIIb
(38.1%)	(61.9%)	(7.3%)	
2	1	3	IV
(66.7%)	(33.3%)	(1.0%)	
**Median score:** **2.0**	**Median score:** **3.0**	**Median score:** **2.5**	**ASA *** (χ^2^, *p* < 0.001)
2	0	2	ASA1
(1.2%)	(0.0%)	(0.7%)	
104	37	141	ASA2
(61.9%)	(31.4%)	(49.3%)	
56	75	131	ASA3
(33.3%)	(63.6%)	(45.8%)	
6	6	12	ASA4
(3.6%)	(5.1%)	(4.2%)	
**Median score:** **4.0**	**Median score:** **7.0**	**Median score:** **5.0**	**CCI *** (χ^2^, *p* < 0.001)
7	0	7	CCI0
(4.2%)	(0.0%)	(2.4%)	
5	0	5	CCI1
(3.0%)	(0.0%)	(1.7%)	
17	1	18	CCI2
(10.1%)	(0.8%)	(6.3%)	
35	1	36	CCI3
(20.8%)	(0.8%)	(12.6%)	
34	8	42	CCI4
(20.2%)	(6.8%)	(14.7%)	
28	12	40	CCI5
(16.7%)	(10.2%)	(14.0%)	
18	30	48	CCI6
(10.7%)	(25.4%)	(16.8%)	
12	28	40	CCI7
(7.1%)	(23.7%)	(14.0%)	
7	17	24	CCI8
(4.2%)	(14.4%)	(8.4%)	
3	10	13	CCI9
(1.8%)	(8.5%)	(4.5%)	
2	8	10	CCI10
(1.2%)	(6.8%)	(3.5%)	
0	1	1	CCI11
(0.0%)	(0.8%)	(0.4%)	
0	1	1	CCI 13
(0.0%)	(0.8%)	(0.4%)	
0	1	1	CCI 15
(0.0%)	(0.8%)	(0.4%)	

**Table 2 jcm-14-06556-t002:** Surgical technique according to age groups; group differences reached statistical significance (χ^2^(5) = 28.593; *p* < 0.001).

Group 1 (<75)	Group 2 (>75)	Total Study Sample	
*N* (%)	*N* (%)	*N* (%)	
5	10	15	**cemented fixation**
(3.0%)	(8.5%)	(5.2%)	
55	9	64	**endofemoral cementless fixation**
(32.7%)	(7.6%)	(22.4%)	
96	86	182	**transfemoral cementless fixation**
(57.1%)	(72.9%)	(63.6%)	
10	12	22	**transfemoral cementless fixation + distal locking**
(6.0%)	(10.2%)	(7.7%)	
1	1	2	**prox. femoral replacement**
(0.6%)	(0.8%)	(0.7%)	
1	0	1	**girdlestone**
(0.6%)	(0.0%)	(0.4%)	
168	118	286	**Total**
(100%)	(100%)	(100%)	

**Table 3 jcm-14-06556-t003:** Microbial analysis according to age groups; group differences did not reach statistically significant differences (χ^2^(3) = 7.554; *p* = 0.056).

Group 1 (<75)	Group 2 (>75)	Total Study Sample	
*N* (%)	*N* (%)	*N* (%)	
109	76	176	**Easy to treat**
(64.9%)	(56.8%)	(61.5%)	
10	11	21	**Difficult to treat**
(6.0%)	(9.3%)	(7.3%)	
37	21	58	**MRS**
(22.0%)	(17.8%)	(20.3%)	
12	19	22	**Culture negative**
(7.1%)	(16.1%)	(7.7%)	
168	118	286	**Total**
(100%)	(100%)	(100%)	

**Table 4 jcm-14-06556-t004:** Mortality and complications associated with cementless septic two-stage revision for group 1 and group 2. Statistically significant differences (*p* < 0.05) are marked (*).

Group 1 Younger 75	Group 2 Older 75	Total Study Sample	
*N*	*N*	*N*	
(%)	(%)	(%)	
1	7	8	mortality * (χ^2^, *p* < 0.001)
(0.6%)	(5.9%)	(2.8%)	
9	5	14	reinfection
(5.4%)	(4.2%)	(4.9%)	
16	10	26	fracture bony flap perioperatively
(9.5%)	(8.5%)	(9.1%)	
6	2	8	fissure perioperatively
(3.6%)	(1.7%)	(2.8%)	
3	0	3	subsidence
(1.8%)	(0.0%)	(1.0%)	

**Table 5 jcm-14-06556-t005:** General complications experienced by each age group; combined complications mean that patients experienced two or more of the named complications. General complications were divided into ischemic (apoplexy and myocardial infarction), thromboembolic (deep vein thrombosis and pulmonary artery embolism), nephrological (renal insufficiency and urinary tract infection), pulmonal (pneumonia), cardial (cardiac decompensation, and endocarditis), hemorrhagic (hematoma, gastrointestinal hemorrhage, and subarachnoid hemorrhage), and soft tissue complications (soft tissue defect, wound healing disorder, necrosis, and erysipelas). Statistically significant differences (*p* < 0.05) are marked (*).

Group 1 (<75)	Group 2 (>75)	Total Study Sample	
*N* (%)	*N* (%)	*N* (%)	
9	4	13	**Dislocation**
(5.4%)	(3.4%)	(4.5%)	
0	3	3	**Ischemic complications**
(0.0%)	(2.5%)	(19.0%)	
3	6	9	**Thromboembolic complications**
(1.8%)	(5.1%)	(3.1%)	
2	5	7	**Nephrological complications**
(1.2%)	(4.2%)	(2.4/%)	
1	1	2	**Pulmonal complications**
(0.6%)	(0.8%)	(0.7%)	
1	7	8	**Cardial complications**
(0.6%)	(5.9%)	(2.8%)	
2	0	2	**Hemorrhagic complications**
(1.2%)	(0.0%)	(0.7%)	
4	3	7	**Soft tissue complications**
(2.4%)	(2.5%)	(2.4%)	
1	6	1	**Postoperative delir**
(0.6%)	(5.1%)	(0.3%)	
0	5	1	**Combined complications 1–9**
(0.0%)	(4.2%)	(0.3%)	
145	78	223	**No complications *** (χ^2^, *p* < 0.001)
(86.3%)	(66.1%)	(78.0%)	
168	118	284	**Total**
(100%)	(100%)	(100%)	

## Data Availability

The data presented in this study are available on request from the corresponding author. The data are not publicly available due to privacy.

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
