# Peer review of "Septic Two-Stage Cementless Hip Revision Arthroplasty Is Safe but Has Higher Complication and Mortality Rates in Older Adults"

_jcm, 2025, doi:10.3390/jcm14186556_

Round 1
Reviewer 1 Report
Comments and Suggestions for Authors
1.The study is described as “retrospective” yet the Methods section does not explain how cases were identified, what inclusion/exclusion criteria were applied, or whether consecutive patients were enrolled. A flow diagram (PRISMA-style) must be added.
2.The mortality analysis is underpowered. With only 8 deaths (2.79 %) overall, the comparison between young (1 death) and elderly (7 deaths) is highly unstable. Provide a post-hoc power calculation or acknowledge this limitation explicitly.
3. Functional outcomes (Harris Hip Score, OHS, EQ-5D) are not reported. Even if not the primary endpoint, some functional data are expected for a contemporary arthroplasty series.
Author Response
Reviewer 1:
Comments and Suggestions for the Authors
- The study is described as “retrospective” yet the Methods section does not explain how cases were identified, what inclusion/ exclusion criteria were applied, or whether consecutive patients were enrolled. A flow diagram (PRISMA-style) must be added.
Answer: Thank you for your comment. The methods section has been revised to provide a clearer and more detailed description of the patient selection process, incorporating the inclusion and exclusion criteria and number of excluded patients
- The mortality analysis is underpowered. With only 8 deaths (2.79%) overall, the comparison between young (1 death) and elderly (7 deaths) is highly unstable. Provide a post-hoc power calculation or acknowledge this limitation explicitly.
Answer: Thank you for your comment. It is true, that comparison of deaths is based on a low sample size. However, Total sample size of this study is highly sufficient and emergence of deaths is a rare event. This may be related to our experience as a tertiary reference center for revision arthroplasty. As recommended we calculated post-hoc power information and added these in the results section, furthermore we added the information of low death rate in the limitations section of the discussion.
- Functional outcomes (Harris Hip Score, OHS, EQ-5D) are not reported. Even if not the primary endpoint, some functional data are expected for a contemporary arthroplasty series.
Reviewer 1:
Comments and Suggestions for the Authors
- The study is described as “retrospective” yet the Methods section does not explain how cases were identified, what inclusion/ exclusion criteria were applied, or whether consecutive patients were enrolled. A flow diagram (PRISMA-style) must be added.
Answer: Thank you for your comment. The methods section has been revised to provide a clearer and more detailed description of the patient selection process, incorporating the inclusion and exclusion criteria and number of excluded patients
- The mortality analysis is underpowered. With only 8 deaths (2.79%) overall, the comparison between young (1 death) and elderly (7 deaths) is highly unstable. Provide a post-hoc power calculation or acknowledge this limitation explicitly.
Answer: Thank you for your comment. It is true, that comparison of deaths is based on a low sample size. However, Total sample size of this study is highly sufficient and emergence of deaths is a rare event. This may be related to our experience as a tertiary reference center for revision arthroplasty. As recommended we calculated post-hoc power information and added these in the results section, furthermore we added the information of low death rate in the limitations section of the discussion.
- Functional outcomes (Harris Hip Score, OHS, EQ-5D) are not reported. Even if not the primary endpoint, some functional data are expected for a contemporary arthroplasty series.
Answer: We acknowledge your advice regarding the assesement of functional outcomes. These were not collected, as this was not the primary aim of our study. However, we have explicitly addressed this aspect as a limitation in the section of the discussion.
: We acknowledge your advice regarding the assesement of functional outcomes. These were not collected, as this was not the primary aim of our study. However, we have explicitly addressed this aspect as a limitation in the section of the discussion.
Reviewer 2 Report
Comments and Suggestions for Authors
The authors present a large cohort of patients undergoing two-stage reimplantation for septic hip revision, analyzing mortality, prosthetic failure, and overall complication rates by stratifying the cohort into two age groups (<75 vs. ≥75 years). They report that while mortality and prosthetic failure rates did not differ significantly between groups, patients ≥75 years experienced a markedly higher rate of general complications. The study is of clear epidemiological interest, given the large sample size and the focus on outcomes in septic two-stage revisions. However, several methodological and statistical aspects warrant revision.
Major Concerns
- Methodology
The methodology section is incomplete and should explicitly describe:
- a) study design, setting, and time frame (retrospective, single-center);
- b) inclusion and exclusion criteria;
- c) detailed treatment protocol (diagnostic work-up, explant and sampling methods, microbial identification/sonication, antibiotic regimen, reimplantation criteria, intraoperative histology/microbiology);
- d) baseline, surgical, and outcome variables collected.
Sample size estimation, statistical method including univariate and multivariate analyses with linear or logistic regression and propensity analysis when appropriate.
- Tables and Baseline Data Presentation
- Move line 83-90 from the methods to the results section.
- Table 1 is poorly informative and should be reformatted. All tables should include three columns: <75 years, ≥75 years, and total, with statistical significance reported.
- Baseline characteristics (sex, age, BMI, interval from index procedure, number of previous procedures, ASA, CCI) should be consolidated in one table with the statistical tests specified.
- If ASA or CCI differ between groups, appropriate adjustments (univariate analysis, regression) are required.
- Radiographic and Surgical Variables
- A separate table should summarize surgical/radiographic details: Paprosky classification, surgical approach, transfemoral access, implant type, cementation (type of cement), cerclages, and microbial flora.
- Lines 94–100 are redundant and should be removed; CCI comparison should be performed with appropriate statistical testing (e.g., Mann–Whitney U).
- Outcome Analysis
- Table 5 should be revised to include statistical significance for each comparison. Post-hoc power analyses should be performed to assess whether non-significant results may be due to limited sample size.
- A further analysis is recommended stratifying patients into “any adverse event” (mortality, reinfection, prosthetic failure, intra/postoperative fracture, general complications) vs. “no event,” with baseline and surgical variables compared. Age should be treated as a continuous variable.
- If multiple baseline/surgical variables differ significantly between groups, multivariate logistic regression should be conducted to identify predictors of poor outcome and to evaluate whether some patients might be better candidates for less aggressive or palliative strategies.
- Discussion
- The unusually low mortality rates compared with registry data require explanation (e.g., type of referral center, case mix, surgical and multidisciplinary team expertise).
- Limitations should acknowledge: - the arbitrary 75-year cut-off, which may obscure age-related distributions; - absence of PROMs or functional outcome measures, which are essential to evaluate whether patients without complications regained functional mobility or remained impaired, raising concerns about overtreatment.
Author Response
Thank you very much for reviewing our paper. All comments are addressed and the changes are with yellow background in the paper.
Comments and Suggestions for the authors:
The authors present a large cohort of patients undergoing two-stage reimplantation for septic hip revision, analyzing mortality, prosthetic failure, and overall complication rates by stratifying the cohort into two age groups (<75 vs. ≥75 years). They report that while mortality and prosthetic failure rates did not differ significantly between groups, patients ≥75 years experienced a markedly higher rate of general complications. The study is of clear epidemiological interest, given the large sample size and the focus on outcomes in septic two-stage revisions. However, several methodological and statistical aspects warrant revision.
Major Concerns
Methodology
The methodology section is incomplete and should explicitly describe:
- study design, setting, and time frame (retrospective, single-center)
- inclusion and exclusion citeria
- detailed treatment protocol (diagnostic work-up, explant and sampling methods, microbial identification/ sonication, antibiotic regimen, reimplantation criteria, intraoperative histology/ microbiology)
- baseline, surgical, and outcome variables collected
Sample size estimation, statistical method including univariate and multivariate analysis with linear or logistic regression and propensity analysis when appropriate.
Answer: Thank you for your comment. We extended the statistical methods where appropriate and specified the treatment protocol and provided a more detailed description of the variables.
Tables and Baseline Data Presentation
- Move line 83-90 from the methods to the results section.
Answer: Thank you for your hint. We shifted this information to the results section.
- Table 1 is poorly informative and should be reformatted. All tables should include three columns: <75years, ≥75 years, and total, with statistical significance reported.
Answer: Thank you for your comment. We reformatted tables 1 and 2 to strengthen the given information. Furthermore, we added for all tables in the results section the column “total study sample”.
- Baseline characteristics (sex, age, BMI, interval from index procedure, number of previous procedures, ASA, CCI) should be consolidated in one table with the statistical tests specified.
Answer: Thank you for your comment. We reformatted tables 1 and 2 to strengthen the given information and added in this table also all baseline information.
- If ASA or CCI differ between groups, appropriate adjustments (univariate analysis, regression) are required.
Answer: Thank you for your comment. Concerning the statistical testing we quite consciously refrained to adjust for differences between groups. As for our goal of this first study, we wanted to see if elderly patients show different reactions and complications rates to the described procedures. As differences in ASA and CCI are indispensably linked to older age, we did not want to adjust for these scales. For forthcoming research your remarks are highly relevant and maybe allow to differentiate more into detail and draw conclusions.
Radiographic and Surgical Variables
- A separate table should summarize surgical / radiographic details: Paprosky classification, surgical approach, transfemoral access, implant type, cementation (type of cement), cerclages, and microbial flora.
Answer: Thank you for your comment. We have provided a more detailed breakdown of the additional regarding surgical approach, implant, cement admixture and cerclage within the text. Presenting these data in tabular form would result in an overly complex table. If the editors request, an additional analysis of the microbial flora and its presentation in tabular form can be performed; however, this would require considerably more time and we believe, that the microbial flora does not have a significant impact on the questions of this study.
- Lines 94-100 are redundant and should be removed. CCI comparison should be performed with appropriate statistical testing (e.g. Mann-Whitney U).
Answer: Thank you. We removed the redundant information and added additionally information for the statistical testing.
Outcome Analysis
- Table 5 should be revised to include statistical significance for each comparison. Post-hoc power analysis should be performed to assess whether non-significant results may be due to limited sample size.
Answer: We added the lacking information of statistical testing and also added post-hoc power analysis in the text. Due to changes of other reviewers comments, table 5 is now numbering “table 3”.
- A further analysis is recommended stratifying patients into “any adverse event” (mortality, reinfection, prosthetic failure, intra/ postoperative fracture, general complications) vs. “no event” with baseline and surgical variables compared. Age should be treated as a continuous variable.
If multiple baseline/ surgical variables differ significantly between groups, multivariate logistic regression should be conducted to identify predictors of poor outcome and to evaluate whether some patients might be better candidates for less aggressive or palliative strategies.
Answer: Thank you for your important hint. As the primary outcome for our study was the specific comparison of two age groups, we abstained from treating age as a continuous variable. Due to the limited number of events, a multivariate analysis would be lacking satisfactory information value. For forthcoming studies with a broader study design and a higher number of participants, for its best in a prospective design we are aiming to integrate more different analysis.
Discussion
- The unusually low mortality rates compared with registry data require explanation (e.g. type of referral center, case mix, surgical and multidisciplinary team expertise).
Answer: Thank you for your comment. In the Discussion, we addressed the markedly lower mortality rate observed in our cohort compared to registry data and attributed this difference to the high level of clinical expertise at our center.
- Limitations should acknowledge: - the arbitrary 75-year cut-off, which may obscure age-related distributions; - absence of PROMs or functional measures, which are essential to evaluate whether patients without complications regained functional mobility or remained impaired, raising concerns about overtreatment.
Answer: Thank you for your comment. We explained and acknowledged the arbitrary 75-year cut-off and the absence of PROMs and functional measures in the limitations of the study.
Round 2
Reviewer 1 Report
Comments and Suggestions for Authors
no comments
Author Response
Thank you very much for reviewing our paper.
No comments were send by reviewer 1.
Therefore no changes are necessary.
Reviewer 2 Report
Comments and Suggestions for Authors
I thank the authors for revising the manuscript according to my previous suggestions. However, some analyses and relevant information are still missing.
Specifically, the statistical significance of baseline variables between the two groups has not been reported (table 1-2). ASA (1-5), CCI (1-15), and Paprosky score (I-IV) — being discrete, ordinal variables — could be presented as median values (particularly CCI), with differences tested using the Mann–Whitney or other suitable non-parametric tests, ideally with input from a statistician. Sex should also be included among the baseline variables.
If ASA, CCI, and Paprosky score are indeed significantly different between the groups, as they appear to be, multivariate analyses and, where appropriate, logistic regression on the selected outcomes (mortality, reinfection, major complications, failure) should be performed to identify independent predictors, considering age as a continuous variable rather than dichotomizing it into groups above or below 75 years. This approach could reveal that age itself is not the main predictor, and that other parameters — particularly ASA, comorbidities, severity of bone loss, and, whenever available, type of microbial flora — may be more reliable predictors of the outcome.
Given the infectious nature of these cases, the type of microbial flora involved should be reported, as certain pathogens (polymicrobial, Gram-negative, antibiotic-resistant, or fungal) are known to be more difficult to eradicate. If such data are unavailable, this should be stated as a limitation.
Finally, if the low mortality and complication rates observed in severe cases — even in geriatric patients — reflect your real-world experience as a multidisciplinary team in a referral center, this should be highlighted. I would recommend concluding that a multidisciplinary approach in a specialized referral center is advisable to optimize outcomes in such complex cases.
Author Response
REVIEWER 2:
I thank the authors for revising the manuscript according to my previous suggestions. However, some analyses and relevant information are still missing.
Answer: Thank you very much for reviewing our paper. The missing points are addressed and the changes and additional text have green background in the new version of the paper.
Specifically, the statistical significance of baseline variables between the two groups has not been reported (table 1-2). ASA (1-5), CCI (1-15), and Paprosky score (I-IV) — being discrete, ordinal variables — could be presented as median values (particularly CCI), with differences tested using the Mann–Whitney or other suitable non-parametric tests, ideally with input from a statistician. Sex should also be included among the baseline variables.
If ASA, CCI, and Paprosky score are indeed significantly different between the groups, as they appear to be, multivariate analyses and, where appropriate, logistic regression on the selected outcomes (mortality, reinfection, major complications, failure) should be performed to identify independent predictors, considering age as a continuous variable rather than dichotomizing it into groups above or below 75 years. This approach could reveal that age itself is not the main predictor, and that other parameters — particularly ASA, comorbidities, severity of bone loss, and, whenever available, type of microbial flora — may be more reliable predictors of the outcome.
Given the infectious nature of these cases, the type of microbial flora involved should be reported, as certain pathogens (polymicrobial, Gram-negative, antibiotic-resistant, or fungal) are known to be more difficult to eradicate. If such data are unavailable, this should be stated as a limitation.
Answer to the Reviewer 2:
As you suggested, we performed multivariate logistic regression analysis with the following variables of interest (dependent variables): one-year mortality, reinfection rate, general complication rate and subsidence. The statistical criteria for performance of a logit regression had been met for all different logit regression analysis: Linearity was tested assessed using the Box-Tidwell (Box & Tidwell, 1962) procedure. Bonferroni-correction was applied to all terms in the model (Tabachnick & Fidell, 2018). All variables were found to follow a linear relationship. Correlations between predictor variables were low (r < .70), indicating that multicollinearity was not a confounding factor in the analysis.
However, for the variable one-year mortality as well as subsidence no significant predictors could be found. Independent variables were reinfection rate, age, Paprosky, ASA, CCI, BMI, diabetes, gender.
For complication rate as well as reinfection the same pattern of results occurred. However, for these two variables, significant regression models could be found, if predictors in the model were reduced and only gender, age, BMI, Paprosky and diabetes were entered in the model. ASA as well as CCI did not enter predictive value:
- Reinfection: The binomial logistic regression model was statistically significant, χ²(8) = 17.439, p = .026, resulting in an acceptable amount of explained variance (Backhaus et al., 2006), as shown by Nagelkerke’s R² = .202. The variable BMI was the only variable, that contributed significantly to the model (p< .001).
- General complications: The binomial logistic regression model was statistically significant, χ²(8) = 27.813, p < .001, however only resulting in low amount of explained variance (Backhaus et al., 2006), as shown by Nagelkerke’s R² = .129. The variables age (p= .016), and diabetes (p= .031) contributed significantly to the model.
Finally, if the low mortality and complication rates observed in severe cases — even in geriatric patients — reflect your real-world experience as a multidisciplinary team in a referral center, this should be highlighted. I would recommend concluding that a multidisciplinary approach in a specialized referral center is advisable to optimize outcomes in such complex cases.
Anwer to the reviewer: This was already mentioned in line 263-265 and is highlighted more in line 266-268.
We hope, we could answer your request to your full satisfaction.
